# Improved Gut Health May Be a Potential Therapeutic Approach for Managing Prediabetes: A Literature Review

**DOI:** 10.3390/biomedicines12061275

**Published:** 2024-06-08

**Authors:** Nosipho Rosebud Dimba, Nhlakanipho Mzimela, Andile Khathi

**Affiliations:** School of Laboratory Medicine and Medical Sciences, College of Health Sciences, University of KwaZulu-Natal, Westville 4000, South Africa; 218006885@stu.ukzn.ac.za (N.R.D.); 218006756@stu.ukzn.ac.za (N.M.)

**Keywords:** type 2 diabetes mellitus, prediabetes, intestinal permeability, gut microbiota, high-calorie diets, zonulin

## Abstract

Given the growing global threat and rising prevalence of type 2 diabetes mellitus (T2DM), addressing this metabolic disease is imperative. T2DM is preceded by prediabetes (PD), an intermediate hyperglycaemia that goes unnoticed for years in patients. Several studies have shown that gut microbial diversity and glucose homeostasis in PD or T2DM patients are affected. Therefore, this review aims to synthesize the existing literature to elucidate the association between high-calorie diets, intestinal permeability and their correlation with PD or T2DM. Moreover, it discusses the beneficial effects of different dietary interventions on improving gut health and glucose metabolism. The primary factor contributing to complications seen in PD or T2DM patients is the chronic consumption of high-calorie diets, which alters the gut microbial composition and increases the translocation of toxic substances from the intestinal lumen into the bloodstream. This causes an increase in inflammatory response that further impairs glucose regulation. Several dietary approaches or interventions have been implemented. However, only a few are currently in use and have shown promising results in improving beneficial microbiomes and glucose metabolism. Therefore, additional well-designed studies are still necessary to thoroughly investigate whether improving gut health using other types of dietary interventions can potentially manage or reverse PD, thereby preventing the onset of T2DM.

## 1. Introduction

Diabetes mellitus (DM) is a metabolic disease characterized by chronic hyperglycaemia due to impaired insulin secretion or insulin action [1,2]. In 2019, approximately 1.5 million individuals died from DM, with type 2 diabetic mellitus (T2DM) causing between 90 and 95% of those deaths [1,3]. T2DM is characterized by hyperglycaemia due to diminished insulin sensitivity [3]. Various complications, including increased intestinal permeability, are seen in T2DM [4]. The onset of T2DM is often preceded by a long-lasting, asymptomatic condition known as prediabetes (PD), which is a state of intermediate hyperglycaemia that occurs between normoglycemia and T2DM [5,6]. This condition is characterized by impaired glucose tolerance or impaired fasting glucose or moderate elevation in glycated haemoglobin [7]. According to a previous study, more than 550 million people will have PD by 2040 [8].

Increased intestinal permeability, also known as “leaky gut,” is a condition in which the lining of the intestines becomes more permeable, allowing translocation of endotoxin lipopolysaccharides (LPSs), peptidoglycan (PG), bacterial fragments, gut bacteria and antimicrobial compounds [4]. When these substances reach the circulation, they trigger an immune response [9]. The current literature indicates that high-calorie diets impact the intestinal mucosal barrier, enterocytes, and overall intestinal permeability, leading to intestinal leakage in type 2 diabetes (T2D) patients [10]. The high-calorie diets cause gut dysbiosis by altering the maintained microbiota diversity in the gut, resulting in a reduction of *Firmicutes* like *Lactobacillus*, *Bacillus*, and *Clostridium*, which protect the intestinal lining [4,11]. A reduction of these microorganisms in the gut causes an increase in zonulin expression [11].

Zonulin is a marker that is associated with a leaky gut and that has been shown in loosened-up tight junctions [12]. The dysregulation of the intestinal barrier causes a constantly elevated inflammatory response that plays a role in many diseases, including diabetes [13]. According to the existing literature, increased intestinal permeability is one of many complications seen in type 2 diabetic individuals [4,14]. In addition to this literature, a recent study conducted in our laboratory showed that this condition begins in the prediabetic state [15]. Therefore, this review aims to review the existing literature on pre-existing mechanisms by which the gut microbiome, in association with diets, contributes to the onset and development of PD and T2D. Furthermore, it outlines possible approaches established in other research studies that have been shown to improve gut health and thereby prevent the progression of PD to T2D.

## 2. The Physiology of Intestinal Permeability

In the stomach, the intestinal tract is lined by a single layer of epithelial cells, which form a part of the dynamic and semi-permeable gut barrier [4]. These cells are highly specialized in allowing specific molecules to pass through the aqueous pores [4]. The smaller molecules use the transcellular pathway, whereas larger molecules use the paracellular system due to size differences [4,9]. Intestinal permeability (IP), an essential property of the intestinal barrier, is formed by tight junction proteins and consists of physical (mucus, epithelial cells), biochemical (bile salts, enzymes, antibacterial proteins), immunological (IgA, immune cells), and microbial components [11,16]. All these factors maintain the proper functioning of the intestinal barrier [11]. The tight junctions also form an essential part of the intestinal barrier, and they are responsible for the adherence of intestinal epithelial cells to one another [17]. These junctions are made up of transmembrane proteins, including adhesions, claudins, tricellulin, coupling adhesion molecules (JAM), angulins, and occluded zonules (ZO) that are attached to the cytoskeleton of the actin cytoskeleton [18,19]. These transmembrane proteins contribute to the structural stability of the intestinal barrier, and they also control the movement of molecules across the cell space within the paracellular pathway [20].

Intestinal epithelial cells function to absorb nutrients while also preventing the translocation of toxins, harmful bacteria (microorganisms) and dietary antigens from the mucosal barrier into the blood circulation [21]. The intestine also consists of what is known as the gut microbiome or microbiota, representing a diversity of microorganisms, including bacteria, fungi, archaea, viruses and eukaryotic cells. They all live inside the intestinal tract in both humans and animals [22]. The intestinal microbiome functions in many processes, such as in the production of crucial vitamins, minerals and nutrients, in the breakdown and absorption of nutrients, prevention of colonization by pathogens, as well as in support of bone development [23]. *Firmicutes* are one of the dominant phyla in the gut, and they function to protect the intestinal barrier, thus preventing inflammation by inhibiting NF-κB and IFN-γ production [24]. They achieve this function by fermenting carbohydrates into various short-chain fatty acids like butyrate [25]. Butyrate functions to maintain the tight junction and strengthen the intestinal barrier [25]. Therefore, it is essential to maintain a balanced gut microbiome composition to promote gut health because an imbalance in the gut microbial diversity or any defect in the intestinal barrier and its components could lead to gut complications such as increased intestinal permeability [26,27].

### Increased Intestinal Permeability (Leaky Gut)

Increased intestinal permeability or “leaky gut” is a condition in which the gut epithelium wall’s tight junctions lose their integrity, allowing increased amounts of toxic compounds and partially undigested food from the lumen to translocate into the bloodstream or into the tissues beneath the gut [13,16]. Various factors change intestinal permeability, and previous studies have shown that it is not only the high-calorie diets or hyperglycaemic state characterized in T2DM patients that affect it, but several other factors such as gut microbiota modification, mucus layer alteration, epithelial damage and other lifestyle choices such as overconsumption of alcohol [28,29]. These factors, in combination with a high-fat high-carbohydrate (HFHC) diet and hyperglycaemia, contribute to even further changes in IP [29,30].

## 3. Markers Associated with Leaky Gut and Microbial Translocation

### 3.1. Zonulin

Zonulin was initially identified as an endogenous human analogue of zonula occludens toxin (Zot), a bacterial enterotoxin produced by the intestinal bacterium Vibrio cholerae [31,32]. Zonulin, a 47 kDa protein, functions to regulate leakiness in the gut by opening and closing spaces between the intestinal lining cells and the tight junctions, to allow the absorption of nutrients and other beneficial molecules into the intestine [12]. The effect of zonulin on increased intestinal permeability is mediated by activation of epidermal growth factor receptor (EGFR) through proteinase-activated receptor 2 (PAR2) as well as G protein-coupled receptor PAR2, which transactivates EGFR [33]. When these two receptors are activated, they reduce transepithelial resistance, which means that intestinal permeability increases [33]. Zonulin loosens the gut barrier integrity, followed by translocation of microbial antigens and endotoxins, triggering an innate and adaptive immune response, causing activation of the proinflammatory cytokines tumor necrosis factor-α (TNF-α), interleukin-6 (IL-6) and interferon-γ (IFN- γ) [12,14]. The activation of these cytokines further damages the intestinal cells and microvilli, resulting in nutrient absorption deficiency [23]. In addition, increased intestinal permeability with elevated zonulin levels has been shown to be associated with inflammatory bowel disease (IBD), celiac disease, multiple sclerosis and diabetes [12,34]. 

Two main factors identified to increase the expression of zonulin in the gut.

#### 3.1.1. Bacterial Colonization

Gut dysbiosis results when there is an imbalance between beneficial and harmful microbial diversity [35]. It is caused by overconsumption of a high-fat diet [23]. In the small and large intestines, the abundance of harmful bacteria increases significantly [35]. Some of these harmful bacteria trigger increased zonulin expression, which weakens the tight junctions as a protective mechanism to reduce or eliminate the bacteria [12]. Symptoms like diarrhoea, constipation, bloating, fatigue and abdominal pain are often associated with an increased zonulin expression, which causes a leaky gut [23]. This allows, even more, larger molecules or toxins to pass through the tight junctions and induces low-grade systemic inflammation that has been shown to worsen the tight junction integrity even further [23].

#### 3.1.2. Gluten

The second factor is gluten and two of its components, gliadin and glutenin, found in rye and wheat, which promote the release of zonulin and cause intestinal diseases [12]. These proteins are insoluble and not easily digested due to high amounts of proline and glutamine, which induce changes in the intestinal barrier, resulting in loss of tight junction integrity [34].

### 3.2. Lipopolysaccharides (LPSs)

LPSs are significant components of the outer membrane of Gram-negative bacteria; they contribute to the structural integrity of these bacteria [9]. LPSs comprise a distal polysaccharide (also termed O-antigen) and a non-repeating “core” oligosaccharide region that is anchored in the outer bacterial membrane by a lipophilic carbohydrate lipid moiety termed lipid A [36]. The gut microbiota consists of most of these endotoxins, kept within bacterial cells and released only after the destruction or replication of the bacteria [36]. LPSs function to protect the membrane from any chemical attacks, induce a robust immune response in humans and increase the cell membrane’s negative charge to assist in stabilizing the entire structure [37]. Under normal circumstances, LPSs are present in the intestinal lumen at low plasma levels when Gram-negative bacteria are destroyed by the immune system [38]. These toxins can be translocated from the lumen into the circulation, but they do not typically affect overall health, because the intestinal barrier is still 90% maintained and intact [7]. However, chronic consumption of a high-calorie or high-fat diet significantly increases LPS levels, a phenomenon often seen in patients with obesity or T2D [14]. In these individuals, elevated LPS levels lead to a leaky gut and other digestive tract diseases [14]. Additionally, they worsen complications such as insulin resistance, chronic kidney disease, and dyslipidemia [14,39]. 

### 3.3. Soluble CD14

Soluble CD14 (sCD14) is a co-receptor for LPSs, and it is an activation marker for monocytes and other mononuclear cells released after stimulation [40]. Upon translocation of LPS from the lumen into the blood, this toxin induces secretion of sCD14 from the immune cells [41]. In a previous study, this marker was used as an indication of LPS exposure in the blood [41]. 

### 3.4. Intestinal Fatty Acid-Binding Protein (IFABP)

Intestinal fatty acid-binding protein (IFABP) is an intracellular protein that is found in the epithelial cells of the mucosal layer of the small and large intestine tissues [42]. According to the literature, high levels of IFABP in the circulation are only translocated when two enterocytes and the mucosal tissue are damaged [43]. Increased serum levels of this protein have been shown to be associated with some common intestinal diseases such as coeliac disease, IBD and irritable bowel syndrome (IRS) [44]. 

### 3.5. C-Reactive Protein (CRP)

Increased intestinal permeability has been shown as one of the risk factors for chronic intestinal diseases because it triggers chronic inflammation [28]. C-reactive protein is an inflammation marker, reported to be associated with zonulin [44]. A previous study investigated patients with T2D who had healthy and unhealthy metabolic profiles [45]. Patients with unhealthy metabolic profiles were shown to have high levels of PG, LPS-binding protein and CRP compared to those with healthy metabolic profiles [45]. It was speculated that these results indicate a certain risk of developing chronic intestinal disease in patients with T2D [45].

## 4. Effect of Obesity on Intestinal Permeability 

Obesity is a chronic progressive disease characterized by an excessive accumulation of adipose tissues (body fats) that increases the risk of health problems [46]. It is a complex disease caused by several factors, including genetics, behavioural choices of diet and exercise and environmental factors [47]. According to Frazier et al. 2021, the large adipose tissue sizes reported in patients with obesity also provide energy for bacterial growth and proliferation [48]. This results in an alteration of the intestinal barrier function, leading to increased intestinal permeability and favouring bacterial endotoxin (LPS) translocation into the circulation [49]. Endotoxemia and overgrowth of adipocytes activate or increase the secretion of pro-inflammatory cytokines, which subsequently cause a chronic low-grade inflammatory state and insulin resistance observed in people with obesity [38].

## 5. Effect of Diet on Intestinal Permeability

The Western diet has also been reported as a risk factor for altering the intestinal barrier and inducing intestinal inflammation [50]. The diet comprises high levels of saturated fat, trans-fatty acid, refined sugars and high-sweetened refined sugars that have been shown to favour the growth of harmful bacteria causing gut dysbiosis [51]. The imbalance of the gut microbiome dysregulates the intestinal barrier, thus causing activation of an immune response upon translocation of the toxins and bacterial antigens [9]. The activation of an inflammatory response causes the secretion of various cytokines, which results in chronic inflammation that damages the enterocytes and epithelial cells of the intestine, causing IBD, celiac disease, multiple sclerosis, and diabetes [30,52,53]. 

## 6. Diabetes Mellitus

DM is classified into two types: type 1 diabetes (T1D), which is an insulin-dependent type, and the most common type, type 2 diabetes (T2D), which is non-insulin-dependent [54]. T1D results from cellular-mediated autoimmune destruction of beta-cells (islets of Langerhans), thereby impairing the release of insulin hormone that is required to make glucose enter cells for energy [1,2,55]. In contrast, T2D is characterized by hyperglycaemia, an increase or high level of blood glucose in the bloodstream due to a defect of the insulin receptor [3]. In T2D, the body cells resist the insulin effect, which drives glucose from the blood into the interior of the cells [56]. Glucose that is not used up by the cells or muscles accumulates in the circulation and further worsens type 2 diabetic complications, such as increased intestinal permeability [10]. The following sections highlight the association between increased intestinal permeability and T2D.

### 6.1. Intestinal Permeability (IP) in T2D

Patients with T2D have long-term hyperglycaemia, which is considered one of the factors that damage the intestinal barrier, epithelial cells and overall intestinal permeability [10]. Chronic consumption of a high-calorie diet has been shown as a secondary factor not only in the progression of T2D but also in the dysregulation of intestinal permeability [29,30]. In the gastrointestinal tract, chronic consumption of a high-calorie or high-fat diet causes an increase in satiety by stimulating the secretion of incretin hormones such as glucagon-like peptide-1 (GLP-1) and cholecystokinin (CCK) [5]. These two hormones limit the rate at which food passes through the digestive tract by delaying gastric emptying and decreasing motility [6]. This prolonged retention of food in the stomach allows the body to break it down into smaller glucose molecules for absorption by the intestinal epithelial cells [57]. The excess glucose that is not used up by the cells enters the bloodstream and exacerbates the hyperglycaemic state in T2D individuals [57]. 

Long-term hyperglycaemia and high levels of saturated fats promote the growth of pathogenic bacteria within the gut [58]. Both these effects shift the maintained equilibrium or imbalances of the gut microbial composition and cause these harmful bacteria to overwhelm and outcompete the essential microbes, resulting in gut dysbiosis [23,35]. Some of the harmful bacteria induce increased zonulin expression, disrupting the tight junctions and leading to the translocation of markers associated with increased intestinal permeability, such as zonulin and LPS, from the mucosal barrier into the bloodstream [14]. All these toxic substances disrupt the protective features of the epithelial cells, resulting in the thickening of the mucus that traps harmful microbial products, destroying epithelial cells, producing an increase in Gram-negative bacteria, causing a decrease in transepithelial resistance and finally disorganizing tight junction proteins, which leads to increased intestinal permeability [11].

### 6.2. LPS Effects in the Circulation in T2D

Hyperglycaemia has been shown to cause changes in the microbiota by shifting the maintained equilibrium of essential and harmful bacteria, resulting in gut dysbiosis, as previously discussed. This dysbiosis has been shown to dysregulate intestinal integrity by increasing zonulin expression, causing increased intestinal permeability that further enhances the translocation of toxins and microbial products, specifically LPS and PG, due to the loss of tight junction proteins such as ZO-1 [12]. LPS is translocated from the intestinal lumen into the circulation by a mechanism facilitated by chylomicrons synthesized from the intestinal epithelial cells in response to an HFHC diet or hyperglycaemic state seen in T2D [4]. LPS has been shown to serve as a ligand for toll-like receptor 4 (TLR-4), which is present on the cell membrane surfaces of immune cells like macrophages, monocytes and others such as endothelial cells and adipocytes [59]. This receptor forms part of the transmembrane pattern-recognition receptor, which provides an essential function in the recognition of microbes and the control of the immune response. Therefore, the binding of LPS to TLR-4 triggers an inflammatory process that results in the release of various cytokines, such as TNF-α, IFN-γ and IL-6, from these immune cells into the lumen [14]. 

TNF-α, during an inflammatory process, augments paracellular permeability by removing transmembrane proteins such as claudin-1 from tight junctions, increasing claudin-2 expression and enhancing occludin degradation [60]. In contrast, IFN-γ induces cytoskeletal rearrangement as well as changes in the tight junction protein expression and localization; this effect increases intestinal permeability [61]. Furthermore, these cytokines in the circulation have been shown to cause peripheral insulin resistance in the liver, muscle and adipose tissues by increasing inflammation and activating JNK1 and NF-κB, which results in serine phosphorylation of insulin receptor substrate-1, leading to insulin resistance [62]. The leakage of LPS also targets the pancreas, triggering inflammation and dysfunction of the pancreatic β cells, in turn causing insulin secretory defects. All these effects further worsen the hyperglycaemic state in T2D [62]. Figure 1 below summarizes the possible mechanisms associated with a high-calorie diet and T2D. The following section will discuss the effects of an HFHC diet on the development of prediabetes and possible complications observed during this state. 

## 7. Prediabetes

Prediabetes is characterized as intermediate hyperglycaemia, in which glucose levels in the blood are above the homeostatic range but below the T2DM threshold [8]. It correlates with the simultaneous presence of insulin resistance and/or pancreatic β cell dysfunction before glucose changes are detectable [63]. According to the American Diabetes Association (ADA), prediabetes criteria are defined by three measurement parameters such as a fasting blood glucose concentrations of 5.6 to 6.9 mmol/L, or oral glucose tolerance test (OGTT) 2-hour glucose concentration of 7.8 to 11.0 mmol/L, or glycated haemoglobin (HbA1c) of 5.7–6.4% [8,64]. The increase in the prevalence of prediabetes has led scientists to predict that more than 470 million people will have prediabetes in 2030 [8]. 

A recent study conducted in our laboratory showed that chronic consumption of an HFHC diet causes the development of prediabetes in male Sprague Dawley rats [15]. In addition, this diet was further associated with changes in intestinal permeability integrity [15]. The findings of this study revealed an increased composition of harmful bacteria and markers associated with a leaky gut, such as zonulin, endotoxin LPS and sCD14 [15]. High levels of zonulin observed were postulated to trigger microbial translocation of endotoxin LPS, which increased secretion of sCD14 and pro-inflammatory cytokines TNF-α, IL-6 and IFN-γ. These cytokines, in response to the translocation, stimulated inflammation, which decreased insulin-stimulated glucose transport and metabolism in adipocytes and skeletal muscles (11). Moreover, pro-inflammatory cytokines caused the activation of nuclear factor kappa B (NF-κB), which in turn activated the serine residue of insulin receptor substrate-1 that caused more inflammation, leading to the development of insulin resistance seen in the prediabetic state. Other than increased intestinal permeability reported by Dimba et al. 2023, other complications, including immune dysfunction, cardiovascular disease and immune response dysregulation, have also been shown to begin in the prediabetic state [65,66]. Hence, approaches to reversing prediabetes have become of interest globally. The following sections will discuss several dietary interventions that have been shown to improve microbial diversity and glucose metabolism in PD or T2D patients.

### 7.1. Dietary Approaches for Gut Health, Prediabetes and T2D Treatment

#### 7.1.1. Banting, Atkins and Ketogenic Diet

Over the past few years, possible approaches to reversing PD have been investigated to reduce or prevent T2D progression and its complications. Dietary interventions, including diets low in carbohydrates and high in unsaturated fat, were reported to manage PD [67,68]. These various types of diets, including the Banting, Atkins and ketogenic diets, have been shown to be beneficial for people with obesity, diabetes, or PD [69,70]. These dietary interventions induce the body to enter a state of ketosis, wherein it becomes accustomed to primarily utilizing fat as its main source of fuel, rather than relying on carbohydrates for sugar [71]. This results in fat loss while preserving muscle mass, thus reducing the secretion of insulin hormone in pancreatic β-cells [72]. The Atkins diet also involves the consumption of a large amount of proteins and restricting carbohydrates to 20 g daily [73]. It starts with a very specific phase and then goes to a maintained phase, which allows a broader range of nutrients to be absorbed by the body [73]. According to the literature, all these types of diets have shown a beneficial effect on regulating blood glucose by improving insulin sensitivity, insulin resistance, and HbA1c levels [74,75,76]. They aid in weight loss by decreasing triglycerides, and they also reduce the risk of cardiovascular diseases by increasing good high-density lipoprotein cholesterol levels [77,78]. All these diets have gained popularity as alternative approaches to reducing the prevalence of PD.

Concerning the microbiota composition, the ketogenic diet in one of the studies was shown to increase the relative abundance of *Akkermansia muciniphila*, which is a producer of SCFAs [79]. SCFAs are a group of fats, primarily generated through the fermentation of dietary fibre by beneficial gut bacteria in the colon, and the most common types include propionate, acetate and butyrate [80,81]. Increased production of these SCFAs significantly strengthen intestinal integrity, modulate insulin resistance and protect against inflammation, thereby potentially reducing the risk of T2D development [79,82]. The postulated mechanism by which SCFAs restore the integrity of the intestine is through enhancing the expression of ZO-1 or by activating a cell signalling pathway that can reduce increased intestinal permeability [83]. However, the effect of the other two diets on improving the microbial composition is still debatable, and additional studies are required for further scientific research. 

#### 7.1.2. Mediterranean Diet (MD)

The MD is a diet composed of rich unrefined carbohydrates, proteins, monounsaturated and polyunsaturated fatty acids and unlimited non-starchy vegetables [84]. In T2D patients, this type of diet has been shown to improve microbial diversity by increasing the bacterial richness of *Bifidobacterium*, *Roseburia* and *Faecalibacteria* [84]. These microbial species function to provide a protective effect in the intestinal barrier and thus help in the digestion of fibres [85]. Moreover, metabolites such as SCFAs were increased, resulting in increased intestinal integrity and insulin sensitivity [84]. A study conducted by Huo et al., 2015 also reported that MD improves glycaemic control (fasting plasma glucose and glycated haemoglobin), body weight and cardiovascular risk factors in T2D patients [86]. Therefore, the MD has been suggested over the years as a potential way to mitigate several health risks concerning PD, T2DM and obesity.

#### 7.1.3. Probiotics and Prebiotics

Probiotics are live microorganisms that in adequate amounts confer a health benefit in a host through increasing beneficial bacteria in the gut [87]. The probiotic bacteria are found in fermented dairy products and foods like kimchi, kombucha, sauerkraut and miso [88]. They have been reported to prevent increased intestinal permeability by maintaining microbiota diversity, improving the integrity of the tight epithelial junctions, and protecting against mucosal injury [89,90]. They achieve all these functions by enhancing the expression of ZO-1 and cell signalling pathways, which limits increased intestinal permeability by also enhancing the junctional complexes [91]. In contrast, prebiotics are a type of dietary fibre that improves gut health by aiding in the digestion of larger oligosaccharide polymers [87]. Prebiotics include inulin, fructooligosaccharides, galactooligosaccharides and lactulose [92,93,94]. They promote the growth of beneficial bacterial species, improve the immune system, keep intestinal integrity intact, and reduce protein fermentation and the presence of pathogenic bacteria [91,92,95]. 

## 8. Conclusions

In summary, our review covered an extensive range of literature demonstrating that high calorie-rich diets induce gut dysbiosis, characterized by an increase in pathogenic bacteria. This alteration increases the expression of zonulin, a protein that disrupts the integrity of tight junctions within the intestinal barrier, facilitating the translocation of endotoxins from the gut mucosa into the bloodstream. Subsequently, these toxins elicit an inflammatory cascade, perpetuating an ongoing immune response that impacts peripheral tissues and beta cells. Ultimately, these cascading effects culminate in insulin resistance, a hallmark of diabetes. However, various types of diets discussed in this review have been shown both to improve gut health and ameliorate PD or T2DM. Hence, these discoveries hold substantial implications for public health, offering vital insights into the necessity of formulating alternative dietary interventions as therapeutic measures. Such strategies aim to enhance or manage gut health, thereby potentially delaying or mitigating the onset and severity of PD.

## Figures and Tables

**Figure 1 biomedicines-12-01275-f001:**
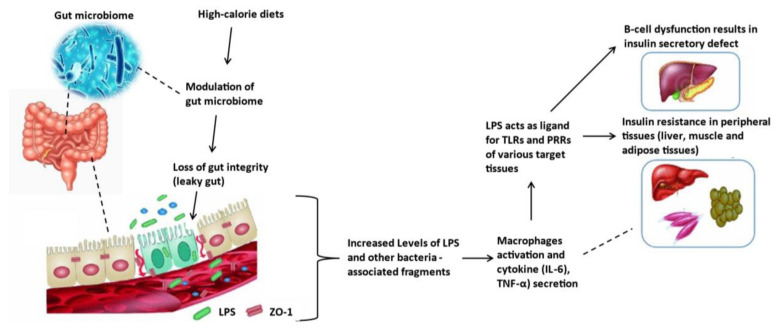
The association of zonulin, lipopolysaccharides, inflammatory markers, and insulin resistance. Adapted from [14].

## Data Availability

Not applicable.

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
