# Peer review of "Improved Gut Health May Be a Potential Therapeutic Approach for Managing Prediabetes: A Literature Review"

_biomedicines, 2024, doi:10.3390/biomedicines12061275_

Round 1

Reviewer 1 Report

Comments and Suggestions for Authors

I would like to thank Dimba et al for their review on the possible role of improved gut health in prediabetes management and diabetes prevention. Many data are gathered in this review. In my opinion, two major drawbacks of this paper are the rather haphazard way these data are presented, and the several points that need to be rephrased or better expressed. More specifically:

Lines 27-28: sentence needs to be rephrased

Lines 40-42: sentence needs to be rephrased

Line 53: “Moreover, to discuss”

After Introduction, a section on how you gathered all these data should follow. It doesn’t have to be a concise description of the methods used but if it stays as it is, it is more of a book chapter than a journal paper.

Lines 62-66: Too long sentence and too complicated. Please try to keep sentences shorter and more concise.

Line 68: “for the adherence”

Line 72: What do you mean by “Interaction of the transmembrane”?

Line 83: How all these features that you describe could “promote better sleep”?

Line 84 and everywhere else: It is IFN-γ or IFN-gamma and not IFN-y

Line 90-91: This sentence should be rephrased to better describe what you want to say

Line 105 and throughout the manuscript: You should first define and then use abbreviations such as HFHC and IP. This holds true for all other abbreviations used throughout the text.

Line 127: a verb is missing

Line 131: increases in the stomach or throughout the gut and especially the small intestine?

Line 134: What do you mean by “leading to the pre-existing state of pathophysiology”?

Line 178: What do you mean by “minor intestinal diseases”?

Line 190: “chronic progressive disease”

Line 202: “comprises”

Line 204: “have been shown to shift the maintained microbiota composition”: sentence must be rephrased

Lines 211-212: No need to describe again what diabetes is, you have already done it earlier in the paper

Lines 230: these two are ANorexigenic hormones

Lines 233-234: sentence needs to be rephrased

Lines 238-240: sentence needs to be rephrased

Lines 276-278: you have already described prediabetes before, no need to refer to it as if you are at the beginning of the manuscript

Line 285: do you have permission for this figure?

In the first paragraph of prediabetes you should maintain mostly the definition of the AmericaN Diabetes Association since it is the one universally accepted

Line 305: you should clarify that your study was on experimental animals

Line 329: “In the body, these diet plans cause the body to”: sentence need to be rephrased.

Lines 360-361: senrence needs to be rephrased

Comments on the Quality of English Language

Extensive language and syntax editing is needed as described above.

Reviewer 2 Report

Comments and Suggestions for Authors

The abstract is an essential component of an article as it provides a concise summary of the study, enabling readers to quickly grasp the key objectives, methods, and outcomes. The abstract should be revised to include main findings/results within the abstract. 

Reviewer 3 Report

Comments and Suggestions for Authors

General comments:

The paper explores the relationship between high-calorie diets, gut health, and the progression of prediabetes to T2DM. Overal an interesting paper evaluating an often undervalued face of prediabetes.

Specific comments:

1. a brief mention of the specific dietary interventions should be discussed to give readers a clear overview.

2. Lines 33 “characterized by impaired glucose tolerance, impaired fasting glucose and moderate elevation in glycated haemoglobin" should be "characterized by impaired glucose tolerance or impaired fasting glucose or moderate elevation in glycated haemoglobin"

3. the role of probiotics and prebiotics in modulating gut microbiota and improving metabolic health should be discussed

4. Mild language polishing is required

Comments on the Quality of English Language

Mild language polishing is required

Round 2

Reviewer 1 Report

Comments and Suggestions for Authors

Thank you, all points raised were sufficiently amended by the authors.